# Activating More Advantageous Neurons Can Improve Adversarial Transferability

## Abstract

Deep Neural Networks (DNNs) are vulnerable to unseen noise, lighting the need to identify the deficiencies of DNNs to mitigate this vulnerability. In the field of adversarial attacks, existing works investigate the deficiencies causing the vulnerability of DNNs, quantifying the vulnerability of DNNs and demonstrating the transferability of adversarial examples where adversarial examples crafted for one model can deceive another. Among the related works, adversarial transferability attracts much attention since transferable adversarial examples enable black-box attacks and raise concerns about DNNs. Although various novel adversarial attacks are presented to improve the adversarial transferability, the property of DNNs that leads to the improvements remains unidentified. This work delves into this issue and reveals that different benign input with different features activates mostly different neurons in a model, and the model may be viewed as an ensemble including different submodels capturing different features. Therefore, an adversarial attack can activate more neurons to generate the adversarial examples, thus probably making the examples applicable to diverse models to enhance the adversarial transferability. Also, data transformation can help exclude wrong answers to boost the adversarial example. The extensive experiments demonstrate the soundness and superiority of our work.

## 1 Introduction

To identify the deficiencies of DNNs, researchers investigate the way to deceive a model by adding noise to inputs, which refers to an adversarial attack. Recently, it reveals that these adversarial attacks can deceive another model while crafting noisy inputs for one model. Thus the transferability study of adversarial attacks is shifted into the highlight and many novel transfer-based adversarial attacks are proposed to improve the transferability of adversarial attacks.

There are various transfer-based adversarial attacks including gradient-based methods (Goodfellow et al., 2014; Kurakin et al., 2018; Dong et al., 2018; Fang et al., 2024), input transformation-based methods (Xie et al., 2019; Zou et al., 2020; Lin et al., 2024; Zhu et al., 2024a), model-related methods (Zhang et al., 2023; Xiaosen et al., 2023; Wang et al., 2024b), ensemble-based methods (Liu et al., 2016; Chen et al., 2023a;b) and generation-based methods (Naseer et al., 2019; Zhu et al., 2024b). Although these methods greatly improve the transferability of adversarial attacks, the deficiencies of DNNs are not clearly identified. Therefore, in this work, we focus on the mechanism of transfer-based adversarial attacks, helping identify the deficiencies of DNNs.

Among these transfer-based adversarial attacks, transformation-based methods are straightforward and popular. These methods improve adversarial transferability by augmenting data and some of these methods take the averaged gradients of several augmented data as the optimization dynamics of adversarial examples. Specifically, given an objective function $J(\cdot)$ and a surrogate classifier $f$, a benign example $x$ and the corresponding label $y$ are taken to generate the adversarial example $x^{adv}$, then the update process of the attacks can be formulated as

$$x_t^{adv} = x_{t-1}^{adv} + \alpha \cdot sign(\sum_i \nabla_{x_{t-1}^{adv}} J(f(\varphi_i(x_{t-1}^{adv})), y)), \tag{1}$$

where $x_t^{adv}$ represents an adversarial example of the $t$-th iteration and the $\alpha$ is the step size. The $\varphi_i$ represents the $i$-th random transformation.

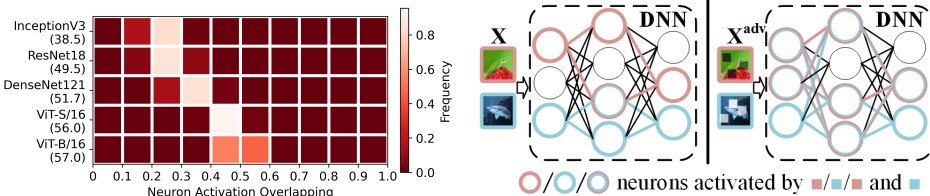

(a) Overlapping of neurons activated by different benign inputs in a model.

(b) Sketch of possible neuron activation in a model.

Figure 1: Neuron activation difference and adversarial transferability of surrogate models. (a) shows the overlapping distribution of neurons activated by different benign inputs in a model and the transfer-based attack success rate (in the "()" below the model name) of different surrogate models. The overlapping is indicated by Averaged Neuron Activation Orthogonality (ANAO) in Eq. 5, illustrating that most of the neurons activated by different inputs are different. Transfer-based attack success rate represents averaged attack success rate over 9 target models. Lower Neuron Activation Orthogonality means more different neurons activated by different inputs. (b) shows possible neuron activation in a surrogate model with benign inputs $X$ and adversarial inputs that have good adversarial transferability, since Figure 1a suggests that the transfer-based attack success rate is higher while different inputs activate more same neurons in a surrogate model.

Given a surrogate classifier $f_\theta^{(s)}(\cdot)$ and a target classifier $f_w(\cdot)$, we take an benign input $x$ into Eq. 1 with $f_\theta^{(s)}(\cdot)$ and $f_w(\cdot)$, respectively. If both the results of the $\sum_i \nabla_{x_{t-1}^{adv}} J(\cdot)$ with the surrogate and target classifier are equal, the process in Eq. 1 is close to the white-box attack, which usually leads to a great attack success rate. Intuitively, the closer the results of the $\sum_i \nabla_{x_{t-1}^{adv}} J(\cdot)$ with the surrogate and target classifier, the better the attack success rate. Thus the introduction of data augmentation to improve adversarial transferability implies that the augmented data may yield results that are closer to the target model's, compared to the original data. This suggests that the neurons activated by augmented data in the surrogate classifier $f_\theta^{(s)}(\cdot)$ are different from the original data, as the objective function $J(\cdot)$ is unchanged.

To observe the neuron activation difference between different inputs, the difference must be quantified in some ways. Thus we start with measuring the activation difference of a classifier $f(\cdot)$ for different inputs in 3.1, and then investigate the mechanism of transfer-based adversarial attacks in the next sections. Finally, based on our findings, an adversarial attack is proposed. This work can be summarized as follows:

- Trained models may be viewed as an ensemble including different submodels capturing different features since the activated neurons of the trained models with different inputs are orthogonal to some extent.

- Data augmentation can help adversarial attacks avoid inefficient perturbations by averaging the gradients of models with several augmented data.

- An adversarial attack is proposed to activate more submodels for improving adversarial transferability and filtering inefficient perturbations by data augmentation.

## 2 RELATED WORK

There are many novel transfer-based adversarial attacks, and we introduce 3 types of related attacks here.

**Input Transformation-Based Attack.** One of the most popular approaches is the input transformation-based attack due to its effectiveness and simplicity. The input transformation-based attack elaborate transformations to enhance adversarial transferability. DIM (Xie et al., 2019) randomly resizes and adds padding to input examples to improve adversarial transferability. Consequently, Zou et al. (2020) calculate the average gradient of several DIM's transformed images to fur-

ther improve adversarial transferability. Then many novel transformations are presented, which calculate the average gradient of the transformed images to improve adversarial transferability. For example, DeCowA (Lin et al., 2024) augments input examples via an elastic deformation, to obtain rich local details of the augmented inputs. L2T (Zhu et al., 2024a) optimizes the input-transformation trajectory along the adversarial iteration, achieving great performance. BSR (Wang et al., 2024a) randomly shuffles and rotates the image blocks to generate adversarial examples.

**Gradient-Based Attack.** This approach elaborates on gradient-based dynamics to improve adversarial transferability. FGSM (Goodfellow et al., 2014) adds a small perturbation in the direction of the gradient, and then I-FGSM (Kurakin et al., 2018) presents an iterative version of FGSM. Consequently, MI-FGSM (Dong et al., 2018) integrates the momentum term into the I-FGSM, as part of the baseline attack. Recently, ADNA (Fang et al., 2024) explicitly characterizes adversarial perturbations from a learned distribution by taking advantage of the asymptotic normality property of stochastic gradient ascent.

**Ensemble-Based Attack.** Different from the other approaches, Liu et al. (2016) presents the first ensemble-based attack which generates adversarial examples using multiple models. Later, several sophisticated ensemble-based attacks are proposed to improve the adversarial transferability. For example, MBA (Li et al., 2023) maximize the average prediction loss on several models obtained by a single run of fine-tuning the surrogate model using Bayes optimization while AdaEA (Chen et al., 2023a) adjust the weights of each surrogate model in ensemble attack using adjustment strategy and reducing conflicts between surrogate models by reducing disparity of gradients of them.

Many of these innovative approaches are experience-based, and thus the mechanisms behind them remain to be further explored.

## 3 METHODOLOGY

To observe the neuron activation difference between different inputs in one way, we try to introduce metrics to quantify the orthogonality of neurons activated by different inputs in a model, and then explore the effect of different inputs on neuron activation in a model, further revealing some relationships between inputs and adversarial transferability.

### 3.1 QUANTIFYING THE ORTHOGONALITY OF NEURONS ACTIVATED BY DIFFERENT INPUTS

The magnitude $|\nabla\theta|$ of the gradient $\nabla\theta$ of the neuron $\theta$ w.r.t objective function can indicate the influence of the weight on the result of a model (Bi et al., 2024), we refer as the extent of neuron activation for the current model in this work. Then we try to formulate metrics to measure the orthogonality of neurons activated by different inputs in a model. Given a model $M_\theta$ with two inputs $x_1$ and $x_2$, we can count the activated neurons in which the $|\nabla\theta|$ is higher than the threshold, and measure the orthogonality between the activated neurons of the model with inputs $x_1$ and $x_2$ by

$$\frac{1}{S} \left\langle \delta(|\nabla\theta_1| - a), \delta(|\nabla\theta_2| - b) \right\rangle, \delta(n) = \left\{ \begin{array}{l} 1, n > 0 \\ 0, n \leq 0 \end{array} \right. , \tag{2}$$

where $S$ is the number of the neurons $\theta$. The $\nabla\theta_1$ and $\nabla\theta_2$ represent the gradients of the neuron $\theta^{(l)}$ of the model $M_\theta$ with the inputs $x_1$ and $x_2$, respectively, while the hyperparameters $a$ and $b$ are the thresholds. The hyperparameter $a$ and $b$ are unequal, due to the incomparable gradient magnitudes of a model with different inputs. To avoid the introduction of the hyperparameters, we try to adopt the $|\nabla\theta|$ as the weight to estimate the orthogonality. However, as shown in Figure 2 Left, the huge size difference between the $\nabla\theta$ of the model with different inputs hinders this process since the model fits different data to different extents for the objective.

Thus, a normalization is introduced into the formulation which can be written as

$$\frac{1}{S} \left\langle \frac{|\nabla\theta_1|}{\sqrt{1/S}\|\nabla\theta_1\|_2}, \frac{|\nabla\theta_2|}{\sqrt{1/S}\|\nabla\theta_2\|_2} \right\rangle . \tag{3}$$

Also, there is another hindrance as shown in Figure 2 Right. There are great size differences between the absolute gradients $\left|\nabla\theta^{(l)}\right|$ of different layers, due perhaps to the property of some structures (e.g.,

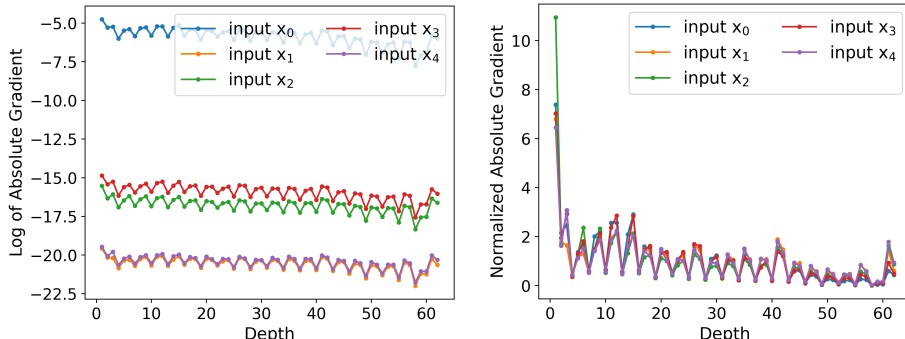

Figure 2: Left: Absolute weight gradients of different layers in a model with different benign inputs. The results are logarithmic due to large numerical differences. Right: Normalized absolute weight gradients (calculated by Eq. 3) of different layers in a model with different benign inputs.

normalization layer). The neurons must be grouped according to the structure and layer they belong to so that the Eq. 3 can make sense. Therefore, we calculate the Eq. 3 with a pair $x_1$ and $x_2$ for one layer to quantify the Neuron Activation Orthogonality (NAO) by

$$NAO(x_1, x_2, l; M_\theta) = \frac{\left\langle \left|\nabla\theta_1^{(l)}\right|, \left|\nabla\theta_2^{(l)}\right|\right\rangle}{\left\|\nabla\theta_1^{(l)}\right\|_2 \left\|\nabla\theta_2^{(l)}\right\|_2} \tag{4}$$

where $\nabla\theta_1^{(l)}$ and $\nabla\theta_2^{(l)}$ represent the gradients of the neuron $\theta^{(l)}$ of the $l$-th layer in the model $M_\theta$ with the inputs $x_1$ and $x_2$, respectively. A lower $NAO(x_1, x_2, l; M_\theta)$ means that the neurons activated by the two inputs are more different, i.e., orthogonal.

We can get a scalar result to compare neuron activation difference between two inputs by Averaged Neuron Activation Orthogonality (ANAO)

$$ANAO(x_1, x_2; M_\theta) = \frac{1}{S}\sum_l S^{(l)} \cdot NAO(x_1, x_2, l; M_\theta), \tag{5}$$

where $S^{(l)}$ is the number of the neurons $\theta^{(l)}$ in the $l$-th layer. Moreover, we sample pairs from a dataset to calculate their ANAOs, observing the reflection of a model on the dataset. Given a model $M_\theta$ and the training set $X \sim \{x_k\}_{k=1}^K$, we calculate the $ANAO(x_i, x_j; M_\theta)$ of different pairs $(x_i, x_j)$ sampled from the training set, and then the distribution of these ANAOs show whether a model $M_\theta$ activates different neurons for different inputs with different features, in other words, whether the model works like an ensemble of multi-models capturing different features.

As shown in Figure 1a, the ANAO distributions of 5 surrogate models suggest that models may work like ensembles of multi-models capturing different features, especially the CNNs. For example, nearly all pairs of data activate less than 30% same neurons in InceptionV3. This implies the model may be viewed as an ensemble composed of some submodels capturing different features and adversarial attacks naturally act like ensemble-based adversarial attacks (Liu et al., 2016), which facilitates the adversarial transferability. Intuitively, we can force examples to activate more neurons to improve adversarial transferability as shown in Figure 1b. Ideally, suppose an example activates all submodels capturing different features. In that case, all the submodels contribute to this adversarial example training. Then the generated adversarial example can attack models including similar one of these submodels.

### 3.2 ACTIVATING MORE NEURONS IMPROVES ADVERSARIAL TRANSFERABILITY

An ideal adversarial example is visually indistinguishable from the original image, and thus perturbation budget $\epsilon$ is introduced as perturbation magnitude limitation. Due to the limitation, adversarial examples pose a challenge in activating all the submodels capturing different features, as shown in Figure 3b. We sample a pair $(x_1, x_2)$ from a dataset as the input of Eq. 5 to calculate ANAO and observe the orthogonality of neurons activated by the data pair $(x_1, x_2)$. If we sample many pairs from

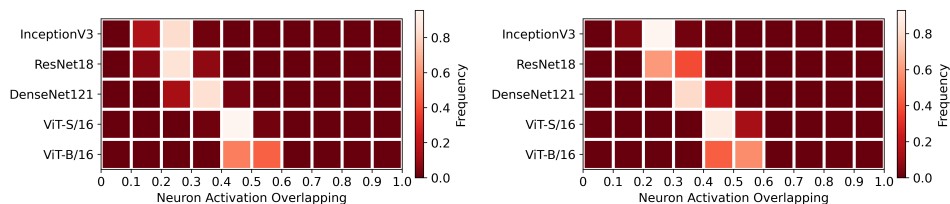

(a) Overlapping of neurons activated by benign inputs in a model.

(b) Overlapping of neurons activated by adversarial inputs (generated by MI-FGSM) in a model.

Figure 3: Neuron activation difference of surrogate models. (a) and (b) shows an overlapping distribution of neurons activated by different benign inputs and adversarial inputs, respectively. The overlapping is indicated by Averaged Neuron Activation Orthogonality (ANAO) in Eq. 5. Lower Neuron Activation Orthogonality means more different neurons activated by different inputs.

the same dataset (ILSVRC2012), then we can count and analyze the frequency where the ANAO of different pairs lie in different ranges. Compare the results in Figure 3b with ones in Figure 3a, given a surrogate model and a gradient-based adversarial attack, the generated adversarial examples can activate more neurons in this model than benign ones, exhibiting to some extent adversarial transferability. This also supports that activating more neurons improves adversarial transferability.

To further demonstrate this, we generate adversarial examples by adding random noise into the benign examples. We evaluate the neuron activation of these adversarial examples by ANAO with a benign example and the corresponding adversarial example as inputs, which can be calculated by where the ANAO can be written as

$$ANAO(x, x^{adv}; M_\theta) = \frac{1}{S} \sum_l S^{(l)} \cdot NAO(x, x^{adv}, l; M_\theta). \tag{6}$$

To avoid bias, we randomly sample the additive noises from a uniform distribution and an image $x_i$ from a dataset $X$, and then calculate the mean ANAO (mANAO) which can written as

$$mANAO(X; M_\theta) = \frac{1}{I} \sum_{x_i \in X} ANAO(x_i, x_i^{adv}; M_\theta), \tag{7}$$

where $I$ is the number of samples. We introduce and evaluate the adversarial transferability by the mean Accuracy. The $I$ is the number of the dataset $X$ and the $S^{(l)}$ is the weight number of the $l$-th layer in the model $M_\theta$, which sums up to $S$. The lower $mANAO$ suggests that the adversarial examples may activate more neurons. Table 1 illustrates the relationship between neuron activation and adversarial transferability, further supporting that, given a specific noise type, activating more neurons can improve adversarial transferability. The results also shows that the noise type has a significant effect on the results, highlight the necessity to identify the effective perturbation type. As such, the next section will be dedicated to do it.

Table 1: The mANAO and mean ASR (Attack Success Rate) of examples with noise. We generate noisy examples as adversarial examples to observe the relationship between neuron activation and adversarial transferability.

| Noise Intensity | | 4 | 8 | 16 | 32 | 64 |
|---|---|---|---|---|---|---|
| Uniform Noise | mANAO | 0.92 | 0.83 | 0.69 | 0.51 | 0.32 |
| | mean ASR | 11.7 | 17.9 | 33.3 | 68.9 | 99.9 |
| Normal Noise | mANAO | 0.97 | 0.91 | 0.81 | 0.66 | 0.43 |
| | mean ASR | 8.0 | 9.8 | 14.9 | 31.2 | 70.9 |

### 3.3 AVERAGING THE GRADIENTS OF AUGMENTED DATA AVOIDS INEFFICIENT PERTURBATIONS

As mentioned in 3.2, there is a perturbation budget $\epsilon$ as perturbation magnitude limitation. To improve adversarial transferability under this limitation, we need to avoid inefficient perturbations and pick more efficient ones instead. Therefore, we discuss this issue in this section.

Data augmentation is widely used to improve data diversity during model training. This technique can help data-driven models to enhance invariance against specific transformation features, and thus the perturbation generated by the gradient of the submodel capturing these features will be ineffective. To improve adversarial transferability, such perturbation should be avoided due to the perturbation intensity limitation. A straightforward solution is to generate an adversarial example by averaging the gradient of this input with random instances of the specific transformation, as this process forces the other submodels to contribute to perturbation updating instead. This is supported by the results in Table 2. Specifically, compared with the baseline with no transformation, the mANAO increases if we take 1 random rotation of the input and optimize the adversarial example for only 1 iteration. This suggests that perturbation generation no longer relies on submodels that capture rotation features. Furthermore, as the number of random transformations increases, the mNAO experiences a decrease, indicating that these transformations facilitate the activation of additional submodels that capture diverse features beyond those related to these transformations. At 10 iterations, random transformations help adversarial examples improve transferability with similar mANAO and perturbation intensity, demonstrating that averaging the gradients of augmented data can avoid inefficient perturbation generation.

Table 2: The role of the used transformation in our proposed AdaAES. The mean perturbation intensity represents the mean of $l_2$-normalization of the generated perturbations. There are just the MI-FGSM with or without the specific transformation during the adversarial example generation.

| Transformation | mANAO | Perturbation intensity | Loss | Mean ASR |
|---|---|---|---|---|
| 1 iteration, 1 random transformation | | | | |
| None | 0.71 | 1.57 | 6.93 | 13.23 |
| Rotation | 0.92 | 1.30 | 0.90 | 7.33 |
| Resized Padding | 0.88 | 1.57 | 1.52 | 10.41 |
| Block Shuffle | 0.84 | 1.57 | 2.63 | 9.18 |
| 1 iteration, 10 random transformation | | | | |
| None | 0.71 | 1.57 | 6.93 | 13.23 |
| Rotation | 0.89 | 1.56 | 1.41 | 9.48 |
| Resized Padding | 0.83 | 1.57 | 2.76 | 14.02 |
| Block Shuffle | 0.80 | 1.57 | 3.67 | 11.50 |
| 10 iterations, 10 random transformation | | | | |
| None | 0.39 | 10.00 | 40.76 | 48.13 |
| Rotation | 0.35 | 10.28 | 16.03 | 73.98 |
| Resized Padding | 0.38 | 10.34 | 19.79 | 77.81 |
| Block Shuffle | 0.34 | 10.17 | 32.94 | 72.34 |

## 3.4 PROPOSED TRANSFER-BASED ADVSERAIAL ATTACK

In this section, we propose an adversarial attack to **Ada**ptively **A**ctivate **E**ffective **S**ubmodels, called AdaAES. Our AdaAES introduces several random transformations to avoid ineffective perturbations and adaptively activate more neurons by calculating the mANAO (Eq. 7) and picking the minimum. The overview and pseudocode of our proposed AdaAES are shown in Figure 4 and 1, respectively.

We first add a tiny additional noise sampled from a uniform distribution into the input, purifying noisy gradients. By default, we make 8 noisy inputs in parallel and then transform these noisy inputs. According to these baseline methods (Dosovitskiy et al., 2020; Simonyan & Zisserman, 2014; Liu et al., 2016), the random rotation, resizing and padding widely used in baseline methods are introduced as part of our transformations ($\varphi_{t-1}(\cdot)$ in Figure 4) due to the reason described in 3.3. Block shuffle is also introduced to suppress the activation of submodels capturing local features, which improves the adversarial transferability for DNNs capturing global features. The hyperparameters of these transformations can be selected automatically by comparing the mANAOs, and thus we only set a large range of the hyperparameters. Concretely, the maximum angle of random rotation is sampled from a uniform distribution (0, 180) by default while the number of split blocks for the block shuffle is randomly sampled from the set $\{1,2,3,4,5\}$. The random resized padding setup follows the setup in Xie et al. (2019), that is, the maximum value of the scaling factor range is uniformly sampled from 1.14 to 1.66 while the minimum is fixed to 1.

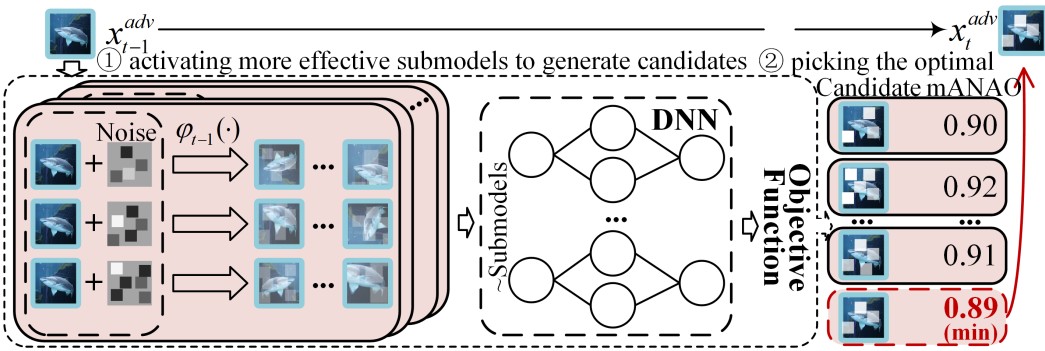

Figure 4: Overview of our proposed AdaAES.

---

**Algorithm 1** Our proposed AdaAES.

---

1: **Input:** an benign example $x_0$, adversarial example $x_t^{adv}$, perturbation budget $\epsilon$, transformation $\varphi$, step size $\alpha$, noise strength $\beta_1$, iteration $T$, candidate number $N_1$, noise number $N_2$, transformation number $I$.

2: Initialize $\alpha = \epsilon/T$, $x_0^{adv} = x_0$, $\overline{g_0} = 0$.

3: **for** $t = 1$ **to** $T$ **do**

4:     **for** $n_1 = 1$ **to** $N_1$ **do**

5:         Sample $noise_{n_2}{}_{n_2=1}^{N_2}$ from the uniform distribution $(-\beta_1, \beta_1)$ and $I$ transformation instances $\{\varphi_t^{(i)}\}_{i=1}^{I}$ with random hyperparamters.

6:         Update the dynamics $g_t = \sum\limits_{n_2}^{N_2} \sum\limits_{i}^{I} \nabla_{x_{t-1}^{adv}} J\big(f(\varphi_i(x_{t-1}^{adv} + noise_{n_2})), y\big)$ .

7:         Update the momentum $\overline{g_t} = \overline{g_{t-1}} + \frac{g_t}{\|g_t\|_1}$.

8:         Generate the candidate $x_t^{(c)} = clip(x_{t-1}^{adv} + \alpha \cdot sign(\overline{g_t}), 0, 1)$.

9:         Calculate the $ANAO(x_0, x_t^{(c)}, f) = \frac{1}{S} \sum\limits_{l} S^{(l)} \cdot \frac{\left\langle \left|\nabla\theta_0^{(l)}\right|, \left|\nabla\theta_t^{(l)}\right| \right\rangle}{\left\|\nabla\theta_0^{(l)}\right\|_2 \left\|\nabla\theta_t^{(l)}\right\|_2}$ (i.e., Eq. 5).

10:     **end for**

11:     Update $x_t^{(adv)}$ with $x_t^{(c)}$ which has the minimum $ANAO(x_0, x_t^{(c)}, f)$ in candidate set $\{x_t^{(c)}\}$.

12: **end for**

---

We use mANAO to show the effect of different numbers of random transformations and the results are shown in Table 3. Table 3 shows that even many random transformations can help activate more neurons. Therefore, we trade off computational cost against performance and set the random number to 160 in total by default.

We repeat the above process 20 times by default in parallel and output 20 candidates. Although more repetitions can lead to performance gains, this also carries a heavy computational burden. We calculate the mANAO of each candidate by Eq. 7 and pick up the candidate with the minimum mANAO which means this candidate can activate more neurons. We repeat these processes for all the iterations and output the generated adversarial example.

## 4 EXPERIMENTS

In this section, we introduce an ablation study to show the role of each component and compare our proposed AdaAES with other attacks, showing the superiority of our method. For fairness, we introduce a widely used PyTorch framework, TransferAttack[1], to train all the transfer-based adversarial attacks in the experiments.

---

[1] https://github.com/Trustworthy-AI-Group/TransferAttack

Table 3: The relationship between the transformation number and the mANAO of the adversarial examples generated by MI-FGSM with the specific transformation number's random transformations (rotation, resized padding, and block shuffle).

| number | 1 | 10 | 40 | 80 | 160 | 320 |
|--------|------|------|------|------|------|------|
| mANAO | 0.4750 | 0.3626 | 0.3281 | 0.3199 | 0.3130 | 0.3101 |

## 4.1 EXPERIMENTAL SETUP

We describe the used dataset, the implementation setup, and the input transformation setup in detail here.

**Dataset.** Following the previous works (Wang et al., 2021; 2023; Zhu et al., 2024a), 1, 000 images are randomly chosen from ILSVRC 2012 validation set (Russakovsky et al., 2015), and these images are classified correctly by the models.

**Implementation Setup.** Following the widely used hyperparameter setup in the works (Dong et al., 2018; Zhu et al., 2024a; Lin et al., 2024), we set the perturbation budget $\epsilon$ to 16/255, iteration number $T$ to 10, step size $\alpha$ to 1.6/255. By default, we adopt noise strength $\beta_1$ as 1.6/255, candidate number $N_1$ as 20, noise number $N_2$ as 8, and Transformation number $I$ as 8.

**Input Transformation Setup.** The input transformation pipeline consists of random rotation, random resized padding, and block shuffle. The hyperparameters of these input transformations are adaptively selected. Random rotation's hyperparameter (i.e., maximum angle) is sampled from a uniform distribution (0, 180) by default. Block shuffle's hyperparameter (i.e., number of split blocks) is randomly sampled from the set $\{1,2,3,4,5\}$. If the number of split blocks is 1, block shuffle is not adopted. Following the setup in Xie et al. (2019), the hyperparameter (i.e., the maximum scaling factor value) of random resized padding is sampled from 1.14 to 1.66 while the minimum is fixed to 1.

## 4.2 ABLATION STUDY

To clearly show the roles of different components of our proposed AdaAES, an ablation study is introduced here and the results are shown in Table 4. Comparing the result of the only transformation component (the $3^{th}$ row in Table 4) with that of baseline (the $1^{th}$ row in Table 4), the results underscore the importance of avoiding ineffective perturbations which greatly enhance the maximum potential performance of the candidate set. Comparing the result of the noise and transformation component (the $4^{th}$ row in Table 4) with that of the complete method, AdaAES (the last row in Table 4), picking the optimal candidate helps yield the optimal result. The additive noise provides a small performance gain in total.

Table 4: Ablation study of our proposed AdaAES. We adopt ResNet18 as the surrogate model here. Cmp. N, T, and C represent the noise, transformation, and candidate components. There is no ablation study of the only candidate component (i.e, "②" in Figure 4) since the candidate component cannot make sense without the random noise and transformation components (i.e, "①" in Figure 4).

| Cmp. | | | Attack success rate (%) | | | | | | | | | | | |
|---|---|---|-------|-------|--------|--------|---------|------|-------|-------|-----|-----------|------|
| N | T | C | Res18 | Res50 | Res101 | NeXt50 | Dense121 | VGG19 | Incv3 | ViT-S | ViT-B | PiT | Visformer | Swin |
| ✗ | ✗ | ✗ | **100.0** | 49.3 | 42.2 | 45.7 | 73.8 | 74.4 | 55.6 | 27.6 | 16.7 | 23.0 | 32.6 | 40.1 |
| ✓ | ✗ | ✗ | 99.9 | 51.1 | 44.5 | 47.3 | 76.8 | 75.7 | 56.0 | 29.0 | 17.1 | 25.1 | 35.7 | 44.2 |
| ✗ | ✓ | ✗ | **100.0** | 92.5 | 91.3 | 93.3 | 99.5 | 99.0 | 97.4 | 82.1 | 63.0 | 67.8 | 83.1 | 83.0 |
| ✓ | ✓ | ✗ | **100.0** | 92.8 | 91.5 | 93.1 | 99.4 | 99.0 | 97.6 | 82.0 | 61.8 | 67.3 | 83.6 | 81.9 |
| ✓ | ✗ | ✓ | 99.9 | 51.2 | 45.0 | 48.7 | 75.8 | 77.1 | 54.9 | 29.9 | 17.3 | 24.2 | 35.0 | 43.7 |
| ✗ | ✓ | ✓ | **100.0** | 93.9 | 92.2 | **93.8** | 99.5 | **99.1** | **98.2** | 82.4 | 62.9 | 66.7 | **84.6** | **84.1** |
| ✓ | ✓ | ✓ | **100.0** | **94.3** | **92.4** | 93.4 | **99.6** | 98.9 | 97.9 | **82.6** | **63.1** | **68.2** | 84.2 | 83.9 |

**The Role of Transformation Number.** We show the correlation between the transformation number and the performance in Figure 5. The results demonstrate that more transformation number can activate more neurons and improve adversarial transferability.

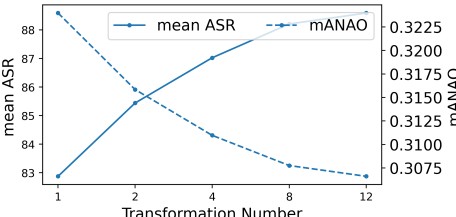

Figure 5: Mean ASR (Attack Success Rate) and mANAO of our proposed AdaAES with different transformation number setups.

### 4.3 COMPARATIVE EXPERIMENTS

In this section, we adopt 5 common neural networks as surrogate models to compare our proposed AdaAES with other advanced attacks and evaluate the attack success rate of different transfer-based adversarial attacks on twelve models including ResNet18 (He et al., 2016), ResNet50 (He et al., 2016), ResNet101 (He et al., 2016), ResNeXt50 (Xie et al., 2017), DenseNet121 (Huang et al., 2017), VGG19 (Simonyan & Zisserman, 2014), InceptionV3 (Szegedy et al., 2017), ViT-S (Dosovitskiy et al., 2020), ViT-B (Dosovitskiy et al., 2020), PiT-B (Zhang et al., 2023), Visformer (Chen et al., 2021), and Swin Transformer (Liu et al., 2021). We pick 7 adversarial attacks as the comparative methods where MI-FGSM and DEM are integrated into our method, and the other advanced methods are proposed recently. Comparison with MI-FGSM and DEM can further show the role of different components in our AdaAES, while comparison with the other advanced methods proposed recently demonstrates the importance of this work in practice.

Table 5: Attack success rate (%) across twelve models on the adversarial examples crafted on ResNet-18 by different attacks.

| Attack | Res18 | Res50 | Res101 | NeXt | Dense | VGG | Inc | ViT-S | ViT-B | PiT | Visformer | Swin |
|---|---|---|---|---|---|---|---|---|---|---|---|---|
| MI-FGSM | **100.0** | 49.3 | 42.2 | 45.7 | 73.8 | 74.4 | 55.6 | 27.6 | 16.7 | 23.0 | 32.6 | 40.1 |
| DEM | **100.0** | 82.5 | 76.8 | 81.8 | 97.5 | 95.1 | 92.1 | 58.7 | 39.1 | 46.0 | 66.3 | 65.9 |
| SIA | **100.0** | 91.9 | 87.6 | 89.7 | 99.2 | 98.6 | 91.5 | 62.7 | 43.9 | 58.5 | 77.3 | 77.0 |
| ANDA | **100.0** | 80.5 | 74.7 | 78.6 | 96.6 | 94.8 | 85.6 | 53.1 | 38.6 | 49.5 | 66.1 | 68.8 |
| BSR | **100.0** | 90.5 | 86.0 | 88.4 | 98.8 | 98.7 | 90.3 | 60.8 | 43.0 | 57.9 | 77.3 | 75.9 |
| DeCowA | **100.0** | 89.0 | 85.0 | 88.3 | 98.5 | 98.4 | 94.4 | 72.3 | 56.5 | 63.7 | 80.5 | 79.8 |
| L2T | **100.0** | 91.5 | 87.6 | 91.6 | 98.6 | 98.8 | 94.8 | 67.4 | 51.0 | 64.7 | 78.8 | 81.2 |
| Ours | **100.0** | **94.3** | **92.4** | **93.4** | **99.6** | **98.9** | **97.9** | **82.6** | **63.1** | **68.2** | **84.2** | **83.9** |

Table 6: Attack success rate (%) across twelve models on the adversarial examples crafted on InceptionV3 by different attacks.

| Attack | Res18 | Res50 | Res101 | NeXt | Dense | VGG | Inc | ViT-S | ViT-B | PiT | Visformer | Swin |
|---|---|---|---|---|---|---|---|---|---|---|---|---|
| MI-FGSM | 47.3 | 30.0 | 28.1 | 28.5 | 44.5 | 47.9 | 97.9 | 23.1 | 13.7 | 16.9 | 24.3 | 28.8 |
| DEM | 77.2 | 57.1 | 55.5 | 57.6 | 78.8 | 76.0 | 99.0 | 47.4 | 30.6 | 35.5 | 47.7 | 49.2 |
| SIA | 87.9 | 69.2 | 65.4 | 69.0 | 85.9 | 83.6 | 99.9 | 49.1 | 34.7 | 46.5 | 58.9 | 61.5 |
| ANDA | 66.1 | 50.1 | 48.4 | 49.8 | 69.5 | 66.0 | 99.7 | 38.1 | 27.2 | 31.8 | 42.9 | 45.6 |
| BSR | 87.7 | 71.9 | 67.5 | 70.6 | 87.0 | 85.6 | 99.8 | 51.1 | 37.0 | 48.7 | 62.8 | 65.6 |
| DeCowA | 78.7 | 57.8 | 57.3 | 61.1 | 78.5 | 78.8 | 98.0 | 47.4 | 32.1 | 38.9 | 49.6 | 54.7 |
| L2T | 83.9 | 70.6 | 67.8 | 70.4 | 84.6 | 80.7 | 98.9 | 52.4 | 37.3 | 49.2 | 56.6 | 61.6 |
| Ours | **92.5** | **75.7** | **73.0** | **76.0** | **92.1** | **89.3** | **99.9** | **65.1** | **45.0** | **53.8** | **66.2** | **70.5** |

Table 7: Attack success rate (%) across twelve models on the adversarial examples crafted on DenseNet121 by different attacks.

| Attack | Res18 | Res50 | Res101 | NeXt | Dense | VGG | Inc | ViT-S | ViT-B | PiT | Visformer | Swin |
|---|---|---|---|---|---|---|---|---|---|---|---|---|
| MI-FGSM | 74.9 | 61.5 | 50.9 | 55.2 | 99.9 | 68.5 | 58.0 | 31.6 | 20.6 | 27.9 | 41.4 | 44.3 |
| DEM | 98.0 | 91.0 | 85.8 | 89.1 | 99.9 | 94.4 | 94.2 | 63.7 | 48.8 | 52.8 | 75.4 | 70.2 |
| SIA | 98.6 | 95.6 | 92.2 | 94.9 | **100.0** | 97.6 | 91.9 | 64.6 | 48.3 | 67.5 | 84.6 | 81.7 |
| ANDA | 93.4 | 86.2 | 81.0 | 83.6 | 99.9 | 89.8 | 82.6 | 53.7 | 40.8 | 55.3 | 71.0 | 69.8 |
| BSR | 98.6 | 95.0 | 89.6 | 93.1 | **100.0** | 97.1 | 88.2 | 62.6 | 49.1 | 66.3 | 83.5 | 79.8 |
| DeCowA | 98.5 | 92.5 | 89.0 | 91.4 | **100.0** | 96.4 | 93.8 | 73.3 | 57.7 | 70.3 | 83.4 | 80.6 |
| L2T | 98.8 | 95.0 | 92.9 | 94.2 | 100.0 | 97.7 | 94.4 | 74.6 | 59.1 | 73.3 | 85.6 | **85.7** |
| Ours | **99.3** | **97.0** | **95.1** | **96.4** | 100.0 | **98.3** | **98.0** | **84.1** | **67.4** | **74.5** | **89.2** | 85.3 |

Table 8: Attack success rate (%) across twelve models on the adversarial examples crafted on ViT-S by different attacks.

| Attack | Res18 | Res50 | Res101 | NeXt | Dense | VGG | Inc | ViT-S | ViT-B | PiT | Visformer | Swin |
|---|---|---|---|---|---|---|---|---|---|---|---|---|
| MI-FGSM | 51.4 | 33.6 | 30.3 | 33.8 | 48.9 | 54.7 | 45.0 | **100.0** | 69.2 | 37.4 | 42.6 | 54.1 |
| DEM | 88.8 | 81.4 | 79.7 | 81.9 | 89.2 | 88.0 | 90.3 | 99.9 | 95.2 | 88.1 | 88.1 | 90.4 |
| SIA | 86.2 | 80.3 | 76.4 | 78.3 | 87.4 | 85.8 | 80.6 | **100.0** | 95.7 | 84.9 | 86.0 | 90.3 |
| ANDA | 70.7 | 60.8 | 57.4 | 60.8 | 73.3 | 71.0 | 67.4 | **100.0** | 89.1 | 67.5 | 69.7 | 77.1 |
| BSR | 87.6 | 82.4 | 82.0 | 83.6 | 89.0 | 87.1 | 84.0 | **100.0** | 94.8 | 90.6 | 88.1 | 91.1 |
| DeCowA | 86.0 | 75.7 | 73.8 | 77.5 | 97.1 | 85.3 | 84.2 | 98.8 | 87.2 | 83.4 | 83.6 | 85.9 |
| L2T | 88.5 | 81.1 | 78.0 | 80.8 | 88.0 | 87.1 | 86.7 | 99.2 | 92.8 | 84.5 | 84.5 | 89.6 |
| Ours | **94.4** | **86.7** | **85.2** | **86.9** | **94.7** | **92.8** | **93.0** | 99.7 | **93.2** | **89.6** | **90.8** | **93.8** |

Table 9: Attack success rate (%) across twelve models on the adversarial examples crafted on ViT-B by different attacks.

| Attack | Res18 | Res50 | Res101 | NeXt | Dense | VGG | Inc | ViT-S | ViT-B | PiT | Visformer | Swin |
|---|---|---|---|---|---|---|---|---|---|---|---|---|
| MI-FGSM | 52.8 | 39.3 | 33.8 | 38.8 | 50.9 | 57.3 | 46.4 | 72.0 | 97.3 | 40.5 | 43.4 | 54.7 |
| DEM | 85.1 | 77.8 | 78.5 | 78.4 | 87.4 | 85.3 | 86.3 | 93.7 | 97.9 | 86.9 | 85.2 | 85.9 |
| SIA | 77.4 | 75.2 | 72.8 | 76.1 | 80.5 | 79.0 | 76.0 | 90.4 | 97.3 | 81.4 | 81.4 | 84.5 |
| ANDA | 67.0 | 60.1 | 58.9 | 60.9 | 70.9 | 69.1 | 66.4 | 84.3 | **97.7** | 66.7 | 68.0 | 73.1 |
| BSR | 74.9 | 73.7 | 71.7 | 73.2 | 78.4 | 75.2 | 75.3 | 84.1 | 93.9 | 78.2 | 76.0 | 79.7 |
| DeCowA | 82.1 | 74.3 | 74.1 | 76.0 | 81.8 | 79.1 | 81.4 | 86.7 | 92.2 | 83.1 | 82.4 | 82.6 |
| L2T | 82.9 | 78.2 | 76.7 | 77.9 | 83.0 | 82.3 | 82.0 | 90.2 | 95.7 | 82.2 | 82.6 | 85.5 |
| Ours | **89.4** | **84.1** | **84.1** | **86.6** | **91.2** | **89.1** | **89.4** | **94.0** | 96.1 | **90.7** | **90.1** | **90.5** |

As shown in Tables 5, 6,7,8 and 9, our proposed AdaAES achieves the state-of-the-art result over the 5 experiments in total, supporting the robustness and superiority of our work.

## 5 CONCLUSIONS

We offer metrics to measure the orthogonality of neurons activated by different inputs, thus investigating the mechanism of transfer-based adversarial attacks and exploring the relationship between inputs, surrogate models, and adversarial transferability from a certain perspective. It reveals that activating more effective submodels in a model can generate better adversarial examples. Activating more neurons can make perturbations effective for more models capturing different features. Averaging the gradients of inputs with random transformation can avoid ineffective perturbation. Also, a straightforward attack based on the above mechanism is proposed to achieve great adversarial transferability.

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
