# OpenReview forum: "Activating More Advantageous Neurons Can Improve Adversarial Transferability"
_ICLR.cc/2025/Conference — ICLR 2025 Conference Withdrawn Submission_

### Official Review · Reviewer_Jv6D · 2024-10-30

**Soundness:** 3
**Presentation:** 2
**Contribution:** 3
**Rating:** 5
**Confidence:** 4

**Summary:**

This paper examines adversarial transferability from the perspective of neuron performance. By introducing the proposed orthogonality measure, it demonstrates how this approach can help eliminate incorrect responses, thereby enhancing the effectiveness of adversarial examples.

**Strengths:**

The motivation is interesting, especially "Thus the introduction of data augmentation to improve adversarial transferability implies that the augmented data may yield results that are closer to the target model’s, compared to the original data. This suggests that the neurons activated by augmented data are different from the original data."

The entire narrative is compelling, and the experimental results validate the effectiveness of the observation.

**Weaknesses:**

1. All the experiments in Section 3 aim to demonstrate the correlation between the ANAO score and ASR, and the authors attempt to use this correlation to explain adversarial transferability. However, while the experiments do show a "correlation," this doesn't necessarily imply that improving the ANAO score would directly enhance adversarial transferability. I would appreciate more theoretical analysis on this point, though I won't lower my rating since the experiments are meant to highlight correlation.

2. Even if orthogonality can be considered an indicator of adversarial transferability, are there any simpler indicators, such as the logit output? For instance, could the values after softmax also provide insight into a sample's influence on adversarial transferability? Could this be considered as neurons in a specialized layer?

3. Why do you use gradients to calculate orthogonality?

**Questions:**

1. The writing could benefit from further refinement. For example, the term "answers" in the abstract is unclear and potentially confusing. Additionally, the "Input Transformation Setup" appears to overlap with the content in Section 3.4, resulting in redundancy. The organization of the experiments is also somewhat confusing—it's unclear why the ablation study is placed in Section 4.2.

2. [1, 2] also explore neuron behavior in adversarial attacks, and I think it would be helpful to mention these works in your related research section. This would help clarify the novelty and contribution of the current paper.

[1] Zhang, Chongzhi, et al. "Interpreting and improving adversarial robustness of deep neural networks with neuron sensitivity." IEEE Transactions on Image Processing 30 (2020): 1291-1304.

[2] Chen, Ruoxi, et al. "NIP: Neuron-level Inverse Perturbation Against Adversarial Attacks." arXiv preprint arXiv:2112.13060 (2021).

---

### Official Review · Reviewer_yzUJ · 2024-10-30

**Soundness:** 3
**Presentation:** 2
**Contribution:** 2
**Rating:** 5
**Confidence:** 5

**Summary:**

This work demonstrates that a pretrained classifier can be considered an ensemble of submodels, each capturing features of different benign inputs. Based on this insight, the authors propose an adversarial attack method, AdaAES, which generates adversarial examples by activating more submodels to improve transferability to unknown classifiers.

**Strengths:**

1. The study investigates the correlation between neuron activations by benign and adversarial inputs within a model.
2. Based on this observation, it introduces a novel method to generate adversarial examples with enhanced transferability.

**Weaknesses:**

1. The paper’s writing could be improved for clarity.

   i)  I found the definition of $\theta$ in Section 3 somewhat confusing. Does it represent a single neuron or all neurons within a layer? Additionally,  $\theta^{(l)}$ is not defined when it appears in line 149.

   ii) The notation ⟨.,.⟩ in Eq. 2 should be defined more explicitly if it represents cosine similarity, and the dimensionality of $\theta$ should be clarified.

   iii) In Table 1, it is unclear whether noise intensity is measured by the $L_2$ or $L_{\infty}$  norm. Specifying this would improve clarity.

   iv) The clip function on line 351 only constrains the candidate within the valid image space; however, it does not account for the perturbation budget, $\epsilon$.


2. Further clarification is needed on why the mANAO value without transformation is lower than with transformation, given 1 iteration and 1 random transformation in Table 2. I had expected that applying transformation while crafting $x^{adv}$ would decrease mANAO and increase the mean ASR. Can you explain this counterintuitive result?

3. According to Table 1, transferability appears to increase with higher noise intensity. Wouldn’t it be expected that as inputs are increasingly deformed by noise, different neurons are activated to reduce confidence in the source samples, thus increasing misclassification? These results do not appear exciting to me.

4. Comparing rows 1, 3, and 5 in Table 4, it seems that including transformations (T) yields good ASR results, while adding N and C to T raises computational costs by $N_1 \times N_2$ with only incremental improvements in ASR. Can you explain good reasons behind the choice of N and C in your method?

5. The comparison with baselines may be not fair, as each iteration of AdaAES calculates gradients $N_1 \times N_2 \times I$
times, might be significantly larger than baselines with the sample number of iterations.  For example, BSR calculates gradients 20 times by transforming the input image 20 times in their default setting. I am curious to see what would happen if BSR averaged the $N_1 \times N_2 \times I$ gradients from that many transformations in each iteration. I like to see a runtime comparison of the proposed method with the baselines.

**Questions:**

1. What is the transformation count 𝐼 for AdaAES? Is it 160 (as per lines 362-363) or 8 (as mentioned on line 397)?
2. In lines 399-400, could you clarify the phrase, “The hyperparameters of these input transformations are adaptively selected”? Which specific hyperparameters are adaptively selected in this context?
3. Fig 3(a) is based on randomly picked benign inputs. I am interested to know the frequency of overlapping neurons if the benign images are sampled from a same class.

---

### Official Review · Reviewer_v2BS · 2024-11-02

**Soundness:** 1
**Presentation:** 1
**Contribution:** 1
**Rating:** 3
**Confidence:** 3

**Summary:**

This paper aims to improve the transferability of adversarial examples in image classification tasks, which could possibly benefit black box adversarial attacks. To achieve this goal, the authors claim to improve the number of activated neurons of the targeted neural network.

**Strengths:**

1. This paper studies an important question relevant to the interest of the community of ICLR.
2. The experimental results demonstrate the proposed method is always close or outperform the strongest baselines.

**Weaknesses:**

The paper is overall poorly organized and the writing is unclear. In detail,
1. The Figure 2 is mentioned before the Figure 1 in the text. Figure 1 is too far away from the context where it is mentioned.
2. An ablation study appears before the major experimental results.
3. The proposed metrics in Section 3.1 lack justification. For example, in Equation 2, the activated neurons of two models $\theta_1$ and $\theta_2$ are compared. However, it didn't clearly stated what kind of models can be compared by Equation 2, e.g. whether they should have the model architecture.
4. As an attack algorithm, the authors also didn't introduce the threat model in detail. For example, in the main proposed algorithm (Algorithm 1), it involves calculate the gradients of a series of models. Do these models include the target model which is used to report the performance in Section 4.3?
5. From the introduction, it is hard to easily get the reason why improve the number of activated neurons can help improve the transferability.

**Questions:**

Plz see the weakness above.

---

### Official Review · Reviewer_vFsF · 2024-11-02

**Soundness:** 3
**Presentation:** 3
**Contribution:** 3
**Rating:** 8
**Confidence:** 3

**Summary:**

The authors introduce a novel metric to measure how different inputs are represented by different parts of the network. Based on this metric, they make an observation that depending on the input, different neurons are activated in the model and adversarial examples typically activate more neurons than clean samples. The latter conclusion partially explains the transferability phenomenon of adversarial attacks. By introducing data augmentation when crafting the attack (a very popular method to improve the transferability), even more neurons are activated according to the new metric which boosts the transferability further. These ideas naturally give way to improve the transferability of existing attacks by picking the adversarial noise that optimizes the given metric thus maximizing neuron activation in the source network.

**Strengths:**

1) The paper is well written and easy to follow.
2) The central idea of using a newly introduced metric to explain and measure the effectiveness of adversarial attack transferability is simple yet effective (especially the reason behind the success of input transformation-based attacks).
3) The claims are well supported empirically.
4) The analysis provided by the authors also naturally introduce a new way to improve the transferability of existing adversarial attacks.

**Weaknesses:**

1) I think Figure 1 (a) and Figure 3 are confusing, and it was hard for me initially to understand them. I would present them in another way or just report a weighted average NAO score for each of the networks.

**Questions:**

1) Is there a way to craft an adversarial attack that would try to activate all of the neurons or submodels to maximize its transferability capacity? Do you have any thoughts on that?
2) Have you looked into measuring the importance of these submodels/neurons? Some of them are probably more important to generate better attacks than others. It would be interesting to see which exactly and why.

---

### Note · Authors · 2024-11-13

**Comment:**

Thanks for the valuable comments, and we will improve the writing.

**Withdrawal Confirmation:**

I have read and agree with the venue's withdrawal policy on behalf of myself and my co-authors.